# Tomography of memory engrams in self-organizing nanowire connectomes

Gianluca Milano [1] ✉, Alessandro Cultrera[2], Luca Boarino[1], Luca Callegaro [2] & Carlo Ricciardi [3] ✉

Self-organizing memristive nanowire connectomes have been exploited for physical (*in materia*) implementation of brain-inspired computing paradigms. Despite having been shown that the emergent behavior relies on weight plasticity at single junction/synapse level and on wiring plasticity involving topological changes, a shift to multiterminal paradigms is needed to unveil dynamics at the network level. Here, we report on tomographical evidence of memory *engrams* (or memory traces) in nanowire connectomes, i.e., physicochemical changes in biological neural substrates supposed to endow the representation of experience stored in the brain. An experimental/modeling approach shows that spatially correlated short-term plasticity effects can turn into long-lasting engram memory patterns inherently related to network topology inhomogeneities. The ability to exploit both encoding and consolidation of information on the same physical substrate would open radically new perspectives for *in materia* computing, while offering to neuroscientists an alternative platform to understand the role of memory in learning and knowledge.

Recent breakthroughs in neuroscience rely on advancements in understanding anatomy and working principles of the human brain cerebral cortex by means of structural and functional imaging. For this purpose, a variety of imaging technologies and mathematical analysis methods have been developed, making possible advancements in mapping the structural and functional connections of our brain, known as connectome. While mapping at the microscale the entire wiring diagram of the human brain composed of $\approx10^{14}$–$10^{15}$ synaptic connections among neurons seems to be unattainable with current technologies, connectomics and network neuroscience have offered a quantitative framework for correlating the activity of macroscale brain regions with specific functions[1–3]. Techniques such as Computed Tomography (CT), Magnetic Resonance Imaging (MRI) and Electroencephalography (EEG) have been explored for macroscale reconstruction of electrical activity and activation patterns in biological brain networks, with the aim of understanding collective dynamics of

human cortical activity[4–8]. Even if the understanding of how our brain works is far from being achieved, it is generally agreed that the ability of our brain to store, retrieve, and process information is intrinsically related to its short-term and long-term memory capabilities arising from the interplay in between structure and function of its neuronal circuits. While short-term memory effects contribute to information processing and computational capability, long-lasting memory effects are at the base of learning and storage of knowledge[9]. In this context, the importance that the network topology has on emerging dynamics and functionalities of brain networks has been highlighted[10]. Importantly, recent advances in neuroscience suggest that memory is stored as learning-induced changes in multiple functionally connected network areas, pointing out that small neuronal ensembles called *engrams* are the basic unit of memory, rather than single synapses[11–13].

In parallel with progresses in neuroscience, advances in artificial neural networks promise to represent a radical change of paradigm in

[1]Advanced Materials Metrology and Life Sciences Division, INRiM (Istituto Nazionale di Ricerca Metrologica), Strada delle Cacce 91, 10135 Torino, Italy. [2]Quantum Metrology and Nanotechnologies Division, INRiM (Istituto Nazionale di Ricerca Metrologica), Strada delle Cacce 91, 10135 Torino, Italy. [3]Department of Applied Science and Technology, Politecnico di Torino, C.so Duca degli Abruzzi 24, 10129 Torino, Italy. ✉e-mail: g.milano@inrim.it; carlo.ricciardi@polito.it

computing, paving the way to neuromorphic computing and artificial intelligence[14]. Trying to fulfill the original goal of neuromorphic computing[15], recent trends rely on the implementation of neural networks on hardware platforms, driving the development of new circuit elements to build artificial neuronal circuits that leverage physics to enhance the computing capabilities[16–20]. With the aim of mimicking biological neuronal circuits where the principle of self-organization regulates both structure and functions, hardware architectures based on self-organized memristive networks of nano objects have attracted growing attention[21–35]. The emerging spatio-temporal dynamics of these artificial connectomes, where an emerging behavior arises from complexity similar to what happens in our brain, make these complex networks versatile physical substrates for hardware implementation of brain-inspired computing paradigms[36–42]. Despite on the one hand devices based on designless nanonetworks have been demonstrated as platforms for hardware implementation of advanced synaptic functions and unconventional computing paradigms, the potentiality of these devices by exploiting their functional connectivity still have to be explored. Indeed, self-organizing memristive networks require a radical change of paradigm in implementing neuromorphic functionalities to take full advantage of their intrinsic multiterminal capability beyond the concept of two-terminal devices, posing at the same time new challenges for characterizing and exploiting their emergent behavior that requires a shift in thinking and designing neuromorphic circuits. In this context, the crucial coexistence of short-term and long-term plasticity (alternatively called *weight* and *wiring* plasticity) on the same physical substrate was just postulated or proved at single unit level (nanowire and nanowire junction)[21]. While visualization of internal dynamics has been proposed through simulations[22,29,32,43], the formation of conductive pathways in nanonetworks has been experimentally investigated by means of passive voltage contrast scanning electron microscopy (SEM) imaging[30,44], conductive atomic force measurements (C-AFM)[44] and thermographic images[28]. While SEM and C-AFM scanning techniques investigate switching phenomena in nanonetworks at the nano/microscale level, lock-in thermography allows to obtain indirect information of main conductive pathways formed where most of the power is dissipated. However, all these techniques do not provide a direct and quantitative information on how the macroscale conductive map of the network, i.e. the internal state of the memristive complex network, evolves under external stimulation.

In this work, we report on evidence of short-term plasticity and long-term memory engrams as changes in the conductivity map of multiterminal nanowire connectomes under electrical stimulation. Going beyond the concept of two-terminal measurements conventionally adopted to characterize memristive cells, we show through a combined experimental and modeling approach that electrical resistance tomography (ERT) allows the investigation of spatially distributed changes in the conductivity distribution across the NW network connectome. We show the tomographical evidence of memory *engram* consolidation through the conversion of short-term changes in the network conductivity map into spatially distributed long-lasting memory traces spanning the connectome. Furthermore, the inherent relationship between spatio-temporal activation patterns and network topology is investigated. By demonstrating the coexistence of spatially distributed short-term and long-term memory effects in the same neuromorphic device, these results represent a radical step ahead towards the development of physical computation that deeply exploits the spatially distributed and temporal dependent signal activity across the nanonetwork, where connection strengths are modulated by inputs history (such as experience in the brain). Moreover, these solid-state devices can represent alternative physical substrates for neuroscientists for implementing new theoretical hypotheses about how memory is formed and recalled in engrams, and how memory is involved with learning in the formation of knowledge.

## Results

### Self-assembled NW connectomes

Self-organizing memristive NW networks were fabricated by drop-casting Ag NWs in solution on $10 \times 10$ mm$^2$ quartz substrates to obtain optically transparent and highly interconnected networks (Methods, Fig. 1a and Supplementary Fig S1). NW density and NW junction density were estimated to be $\sim 10^5$ mm$^{-2}$ and $\sim 10^6$ mm$^{-2}$, respectively (Supplementary Note 1). The emergent memristive behavior of the network arises from the interaction of a multitude of memristive NW junctions, where the switching mechanism relies either on the formation of an Ag conductive filament across the insulating polyvinylpyrrolidone (PVP) shell layer connecting the metallic cores under the action of an applied electric field (Fig. 1b) and/or on electrical-induced modification of the network structure (Supplementary Note 2)[21].

### Spatio-temporal information processing with multiterminal NW networks

A schematization of the experimental setup used for multiterminal characterization of the emergent memristive behavior is reported in Fig. 1c, where the NW network is connected to the experimental setup by means of 16 needle probes (neuron terminals) wired to a switching matrix that allows reconfigurable wiring configurations (Methods, Supplementary Fig S2). An example of the network response to the spatio-temporal stimulation pattern reported in Fig. 1d composed of pulse trains (temporal domain) each applied to different pairs of neuron terminals (spatial domain) is reported in Fig. 1e. The stimulation of a pair of network terminals with a voltage pulse results in a potentiation of the corresponding synaptic pathway with enhanced effective conductance followed by a spontaneous relaxation where the conductance progressively relaxes over time towards a new conductance state (refer for example to pulse stimulation and network response in black time traces in Fig. 1d, e, respectively). A comparison of the output characteristics of NW networks with nominally identical features is reported in Supplementary Fig S3. While spontaneous relaxation, which gives rise to short-term synaptic plasticity effects, is related to volatile switching effects in memristive network elements, long-lasting effects rely on changes of the network topology[21]. Importantly, temporally correlated pulses applied within a short time interval result in a gradual increase of the effective synaptic pathway conductance, emulating paired pulse facilitation (PPF) of biological synapses by exploiting the competing effects of memory enhancement and spontaneous decay (refer to red and blue time traces in Fig. 1d, e, respectively). Because of the functional connectivity, the system endows heterosynaptic plasticity, meaning that changes in the synaptic pathway conductance can be observed not only in correspondence with direct stimulation of the synaptic pathway (circled in Fig. 1e), but also in non-directly stimulated synaptic pathways. In particular, the heterosynaptic effect depends on the spatial location of neuron terminals, where the effective conductance of a synaptic pathway is not strongly affected by stimulation of peripheral neuron terminals (details in Supplementary Fig S4).

### Mapping self-organizing connectomes

The above-discussed synaptic plasticity effects arise from a memristive reconfiguration of the network, where conductance dynamics in between selected neuron terminals depend both on their spatial location and on peculiar dynamics of the NW connectome under spatio-temporal stimulation. In this context, two-terminal characteristics of these multiterminal devices (as reported in Fig. 1d, e) are only local manifestations of hidden changes in the spatio-temporal distribution of conductivity across the whole network connectome. We experimentally unveil hidden network dynamics by electrical resistance tomography, which allows a quantitative imaging of both local conduction properties and emerging behavior of the NW connectome at the macroscale. This non-scanning technique is based on the

reconstruction of the spatial distribution of conductivity across the network from boundary electrical measurements. The set of four-terminal resistance measurements required for ERT reconstruction was experimentally acquired through an adjacent pattern measurement scheme (details of the measurement protocol in Supplementary Note 3)[45,46], by injecting current in between a pair of adjacent terminals through a constant applied voltage bias ($V_{source}$) while measuring voltage across remaining pairs of adjacent terminals ($V_{sense}$), as schematized in Fig. 2a (details in Methods, Supplementary Fig S5). While maximizing the signal-to-noise ratio, the measurement protocol prevents the onset of sample alterations (Supplementary Fig S6)[45]. Simulations were performed by modeling the NW network as a memristive grid-graph[36,47] (details in Methods). This modeling approach relies on the approximation of homogeneous and high-density NW network as a continuous medium, followed by parcellation of the 2D domain and approximation as a regular grid of memristive devices, as schematized

in Fig. 2b (details in Supplementary Note 4 and Supplementary Table 1). Here, each memristive edge of the grid-graph represents the memristive interaction among different areas (nodes) of the network composed of a multitude of NWs. It is worth mentioning that a similar approach based on parcellation is exploited in neuroscience where, due to the difficulties of mapping the entire connectome at the single synapse level, the graph representation of the brain connectome relies on describing entire brain regions as nodes, where edges provide connections in between these regions[3].

Transresistance patterns, consisting in a set of $n(n-3)$ measurements (i.e., 208 measurements with $n = 16$ terminals), of an experimental and grid-graph modeled nearly homogeneous NW network in the pristine state are reported in Fig. 2c. Here, patterns are compared also with the ones simulated for a perfectly homogeneous and continuous sample (details in Supplementary Fig S7). Despite the microscopic structure and randomness of the NW network, main features of the experimental

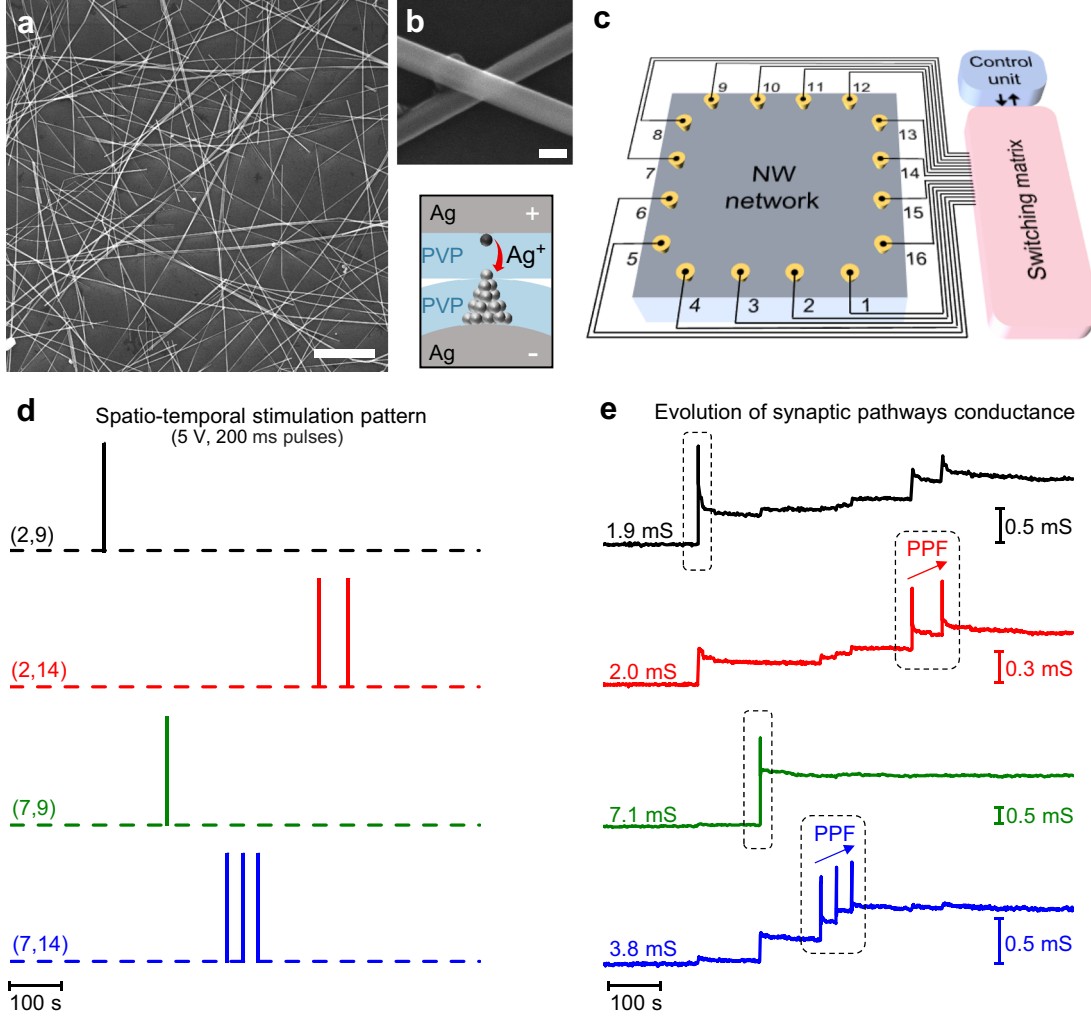

**Fig. 1 | Multiterminal memristive NW networks for spatio-temporal information processing. a** Representative SEM image of an Ag NW network (scale bar, 5 μm). **b** Detailed SEM image of the intersection in between two nanowires (NW junction) acting as an electrochemical memristive cell (scale bar, 100 nm) and schematization of its working principles. The switching mechanism is related to the formation/rupture of a metallic conductive filament across the insulating PVP NW shell layer connecting the two Ag NW cores under the action of the applied electric field. **c** Schematization of the multiterminal memristive NW network device and control system. A 10 × 10 mm² NW network is contacted by means of needle probes and connected to a switching matrix driven by a control unit acting as neuron terminals. The effective conductance in between two neuron terminals represent the weight

of the corresponding synaptic pathway. **d** Example of a spatio-temporal stimulation pattern composed of different pulse trains (temporal domain) (pulses of 5 V, 200 ms) each applied to different pairs of neuron terminals (spatial domain) and **e** corresponding evolution of the effective conductance of synaptic pathways. In (**d**), each channel corresponds to the stimulation of the pair of terminals in parentheses. When not stimulated, electrode terminals were left floating (dashed line). In (**e**), circled areas show synaptic plasticity changes related to direct stimulation of the corresponding synaptic pathway, where temporally correlated pulses applied within a short interval result in paired pulse facilitation (PPF). The effective conductance of each pair of neuron terminals was sequentially measured with a read voltage of 10 mV. During stimulation pulses, the conductance of the directly stimulated synaptic pathway was continuously recorded.

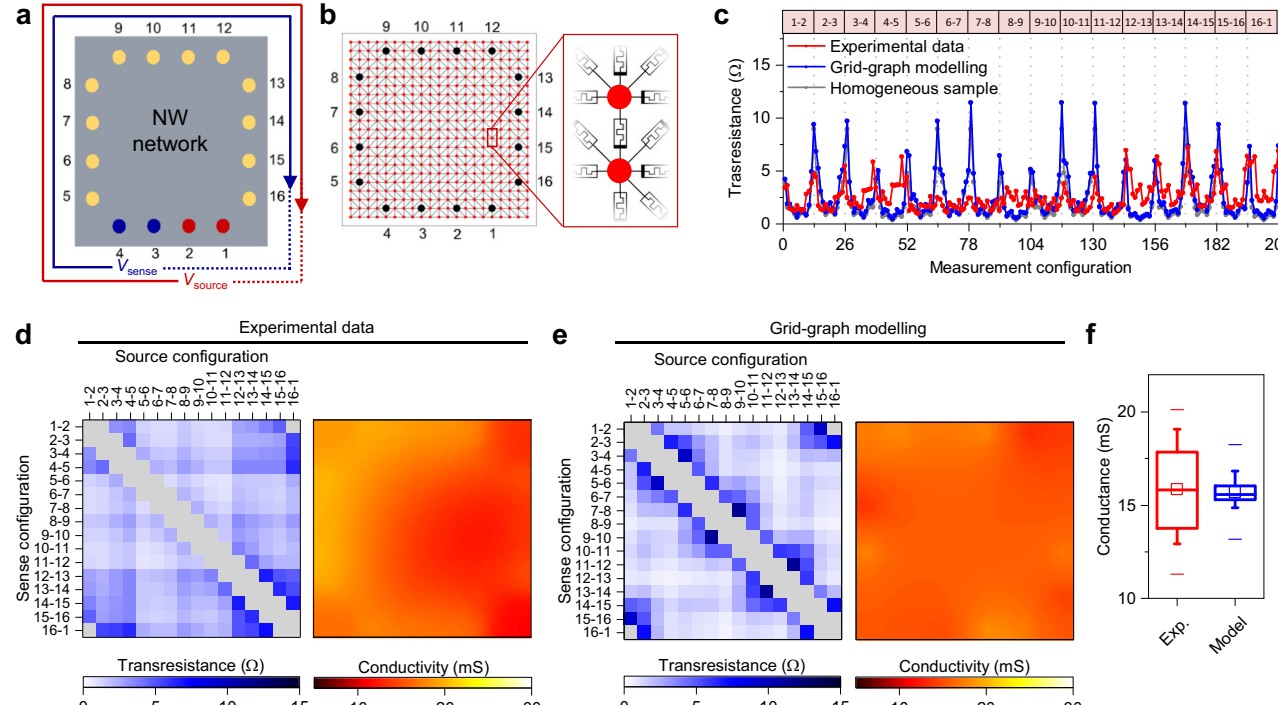

**Fig. 2 | Experimental and modeling electrical imaging of homogeneous NW networks. a** Schematic representation of the adjacent measurement protocol where a source voltage ($V_{source}$) is applied between a pair of adjacent terminals (e.g., terminals 1–2) while measuring the voltage ($V_{sense}$) difference across a pair of different terminals (e.g., terminals 3–4). For each fixed pair of source terminals, the voltage drop is measured in between other adjacent pair of terminals with a counterclockwise sequential scheme (13 sense configurations, avoiding 2-terminals and 3-terminal measurements). The same measurement scheme is repeated by shifting counterclockwise the source adjacent pair of terminals (16 source configurations), obtaining a set of 13 × 16 = 208 transresistance values. **b** Grid-graph model where the NW network is represented by a grid-graph with random diagonals where graph nodes are connected by memristive edges. The network electrode terminals (black nodes) are placed according to the geometry of the experimental setup. **c** Transresistance patterns obtained with the adjacent protocol scheme of an experimental NW network, a grid-graph modeled network and a simulated homogeneous sample with a median conductivity of 15.7 mS. The source configuration is labeled at the top. Impedance matrices (obtained from measurement pattern reported in panel c) and corresponding conductivity maps obtained by ERT reconstructions from (**d**) experimental data and (**e**). grid-graph modeling of NW networks. **f** Quantitative comparison of conductivity values obtained from ERT reconstruction of experimental data and grid-graph modeling, where box plots represent the distribution of conductivity pixels in maps reported in (**d**, **e**). Midlines represent median values, squares the mean values, boxes the 25th and 75th percentiles, whiskers the 10th and 90th percentiles, and lines the maximum and minimum values.

transresistance pattern obtained on a high density and nearly homogeneous NW network are in good agreement with the pattern obtained from the grid-graph model. Also, the good agreement of main features with the simulated homogeneous sample's pattern supports the approximation of the high density and homogeneous NW network as a 2D (continuous) memristive material at the nanoscale.

The transresistance pattern can be rearranged to represent part of the (indefinite) impedance matrix $Z$ that endows the electrical representation of the multiterminal network (Supplementary Fig S8). Together with the sample's geometry and position of network terminals, the impedance matrix is then passed as input for ERT reconstruction of the network conductivity map by means of optimization techniques (details in Methods). This involves solving an inverse problem, i.e., determining the conductivity map corresponding to a given impedance matrix, where the spatial resolution relies on the amount of available boundary information. In the case of the here reported 16-contacts ERT setup, the spatial resolution can be considered in the order of the contact distance (≈2 mm) (details of spatial resolution and traceability in Supplementary Note 5 and Supplementary Fig S9). Note that this approach shares similar basic principles exploited for solving the inverse problem of EEG, where the location of brain areas generating neuronal activity can be inferred from EEG potential measurements on the scalp[7]. Fig. 2d, e reports impedance matrices (obtained from measurement patterns reported in Fig. 2c) and corresponding conductivity maps of an experimental sample and grid-

graph modeled NW network, respectively (parameters of the grid-graph modeling were retrieved from interpolation of experimental data, details in Supplementary Fig S10). Since the considered NW networks fulfill the reciprocity theorem of linear passive electrical networks, matrices $Z$ are symmetric (Supplementary Note 6, Supplementary Fig S11). A substantially homogeneous conductivity map was observed in the case of the experimental NW network, in good agreement with results obtained from ERT reconstruction of the grid-graph model. A quantitative comparison of the conductivity distribution of ERT map pixels is reported in Fig. 2f, where it is possible to observe, despite the high uniformity of the NW network, a wider distribution of conductivity values in the experimental network due to the presence of local variations of conductivity related to the randomicity of the deposition process. Further details on long-term stability of neuromorphic NW networks are reported in Supplementary Fig S12.

## Tomography of memorizing-forgetting effects
The emergent dynamics of the functional synaptic conductivity map of the NW connectome under spatio-temporal stimulations is assessed by acquiring the impedance matrices of the system over time, as conceptually schematized in Fig. 3a. Then, the acquired dataset is used as input for the ERT reconstruction algorithm to map the dynamic evolution of the network conductivity map (Methods). Figure 3b shows a typical experimental temporal evolution of the synaptic effective conductance in between a pair of neuron terminals (6 and 15) in the

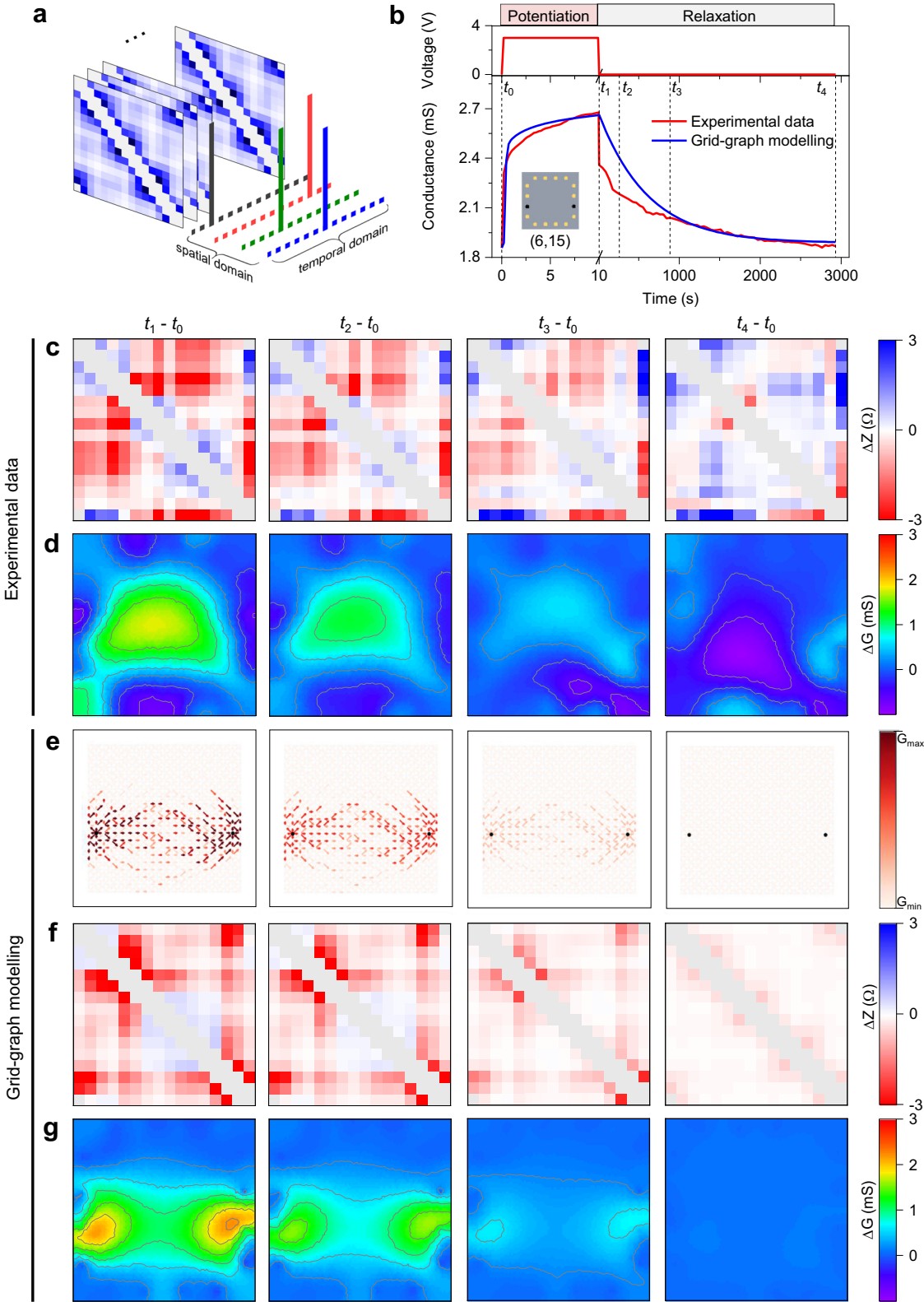

homogeneous network, where stimulation was tailored to induce short-term synaptic plasticity (Supplementary Note 7). Under these circumstances, the effective conductance shows potentiation with an increase of the effective conductance during direct stimulation of selected terminals with a voltage pulse and subsequent spontaneous relaxation after the end of stimulation towards the pristine ground state. As it can be observed, the effective conductance time trace of the

NW network working in the short-term memory regime can be well interpolated by the grid-graph model[36,47] where memristive dynamics are regulated through a potentiation-depression rate balance equation[48] (details in Methods and Supplementary Note 8). At the same time, impedance matrices were acquired over time in both experiment and grid-graph modeling. Figure 3c reports the experimental differential impedance matrices $\Delta Z$ calculated as the difference

**Fig. 3 | Mapping dynamic evolution of activation patterns in a homogeneous memristive nanowire network connectome. a** Conceptual schematic representation of time-dependent acquisition of impedance matrices in multiterminal memristive networks during spatio-temporal stimulation of the system for mapping the spatio-temporal evolution of network conductivity. **b** Experimental and simulated evolution of the effective conductance in between a pair of stimulated terminals under direct stimulation followed by spontaneous relaxation. Potentiation was performed by applying a voltage pulse (3 V, 10 s) in between terminals 6 and 15, while the relaxation was monitored over time in between the same terminals with a read voltage of 10 mV. The inset shows the position of the stimulated terminals. **c** Experimental differential impedance matrices acquired over time (head and tail levels have been assigned to maximum and minimum levels, respectively, for better data visualization) and (**d**) corresponding experimental

differential conductivity maps by ERT reconstruction of the NW network during stimulation reported in (**b**). **e** Evolution of the memristive network by grid-graph modeling (**f**) corresponding simulated differential impedance matrices obtained from the grid-graph model and (**g**) corresponding simulated differential conductivity maps by ERT reconstruction during stimulation reported in panel b. Differential impedance matrices were obtained by measuring the differential transresistance values ($\Delta Z$) between a selected timestamp and $t_0$ (before stimulation). Differential conductivity maps were obtained by evaluating the local variation of conductivity ($\Delta Z$) through differential map reconstruction. Differential matrices and conductivity maps correspond to timesteps labeled in (**b**). Experimental and modeled ERT map reconstruction evidenced the emergence of a conductive pathway connecting stimulated terminals that progressively vanishes during spontaneous relaxation.

of entries of $Z_i$ at a selected timestep $t_i$ and entries of $Z_0$ at $t_0$ (i.e., the pre-stimulation impedance matrix), while Fig. 3d shows the corresponding reconstructed differential conductivity maps (details in Methods). Dynamics of the experimental effective conductance, impedance matrices and differential conductivity maps evolution are reported in Supplementary Movie 1. Figure 3e shows the spatio-temporal evolution of conductance by means of the grid-graph modeling, Fig. 3f the differential impedance matrices obtained from modeling and Fig. 3g the corresponding reconstructed differential conductivity maps. Dynamics of simulated effective conductance, grid-graph modeling, impedance matrices and differential conductivity maps are reported in Supplementary Movie 2. Differential impedance matrices after stimulation show the emergence of peculiar features that reflect local changes of conductivity across the NW network, where main emergent features of the experimental matrix are captured by the matrix obtained from the grid-graph model (refer to panels c and f of Fig. 3). The peculiar features emerging in $\Delta Z$ after stimulation reflects the emergence of an activation pattern (memory trace) with enhanced conductivity in the network connectome that can be observed in the experimental and simulated topographical reconstruction of the differential conductivity maps $t_1 - t_0$. Note that the growth dynamics of the memory trace is regulated by the local electric potential and proceeds from the stimulated terminals, as detailed in Supplementary Fig S13. Simulations and experiments reported in Fig. 3 suggest that electrical stimulation does not result only in the formation of a single pathway with enhanced conductivity in correspondence with the shortest path connecting the stimulated contacts. Conversely, activation patterns that are distributed across the network clearly arise due to the occurrence of conductive pathways with multiple branches. These results are in accordance with experimental studies by passive voltage contrast SEM imaging[30] and with simulations reported in Supplementary Fig S14, both showing the formation at micro/nanoscale of conductive pathways composed of multiple branches. After stimulation, the impedance matrix of the system tends to relax to the ground state due to the short-term memory of the network connectome, as testified by the progressive reduction of the differential transresistance values over time in both experimental and simulated data. The fading memory (relaxation) of transresistance values of the system reflects the progressive vanishing of the memory trace, as can be observed by tomographic reconstruction of differential conductivity maps over time for $t > t_1$, where the initial distribution of conductivity is nearly restored at $t = t_4$. Results show qualitative agreement between experimental and simulated conductivity maps and their spatio-temporal dynamics when high density and nearly homogeneous NW networks are considered.

### Short-term plasticity, memory consolidation and network topology

While high-density networks with nearly homogeneous distribution of NWs can be modeled as a continuous and uniform memristive

2D-like material, a complex emergent behavior arises in non-homogeneous connectomes, due to the interplay in between emergent functionalities and the peculiar network structure. An example of the conductivity map of a non-homogeneous NW network in its pristine state is reported in Fig. 4a, showing higher conductivity at the bottom-left corner (Supplementary Fig S15). Nonuniform conductivity across the network in pristine state is inherently related to the local density and topology of intersecting NWs (Supplementary Note 9)[46]. The emergent behavior of the non-homogenous connectome was assessed by means of voltage pulse stimulations applied in between selected neuron terminals (terminals 6 and 13). While stimulation with a low voltage pulse results in a subsequent relaxation of the network towards the ground state over time (Fig. 4b), stimulation with higher voltage pulses results in a stronger potentiation characterized by long-lasting changes in the effective conductance in between stimulated terminals (Fig. 4c). Figure 4d, e reports differential impedance matrices and corresponding reconstructed differential conductivity maps, respectively, showing the evolution of the non-homogeneous connectome after stimulation with 1 V pulse (Supplementary Movie 3). Differential conductivity maps after stimulation ($t_{A1} - t_{A0}$) evidenced the formation of enhanced conductivity areas near stimulating terminals, with a larger stimulated area near terminal 6 with respect to terminal 13. In this case, the asymmetry of the spatial activation pattern is inherently related to the network topology. Indeed, the higher conductivity in the pristine state of the area near terminal 6 results in a nearly equipotential area surrounding terminal 6 during stimulation, with the voltage drop that occurs mainly at the boundaries in between this area (that act as a nearly equipotential virtual electrode) and the neighbor areas with a lower conductive pristine state (details in Supplementary Fig S16). Thus, the non-homogeneous initial distribution of conductivity across the network related to its topology is responsible for a peculiar redistribution of the electrical potential that drives switching events across the network during electrical stimulation, resulting in the formation of peculiar morphologies of spatio-temporal memory traces. Reflecting the relaxation towards the ground state of both the two-terminal effective conductivity and of the differential impedance matrices, a progressive vanishing of these activation patterns can be observed in the connectome over time due to short-term memory characteristics of the network. Differential impedance matrices and corresponding reconstructed differential conductivity maps after stimulation with a 2 V pulse are reported in Fig. 4f and g, respectively (Supplementary Movie 4). As can be observed, a stronger stimulation results in a larger activation pattern across the connectome with enhanced conductivity with respect to the activation pattern generated by stimulation with a lower amplitude pulse. In this case, long-lasting changes in impedance matrices result in activation areas across the reconstructed conductivity map that hardly relax over time, indicating that the transition from short-term to long-term memory can be achieved through appropriate electrical stimulations.

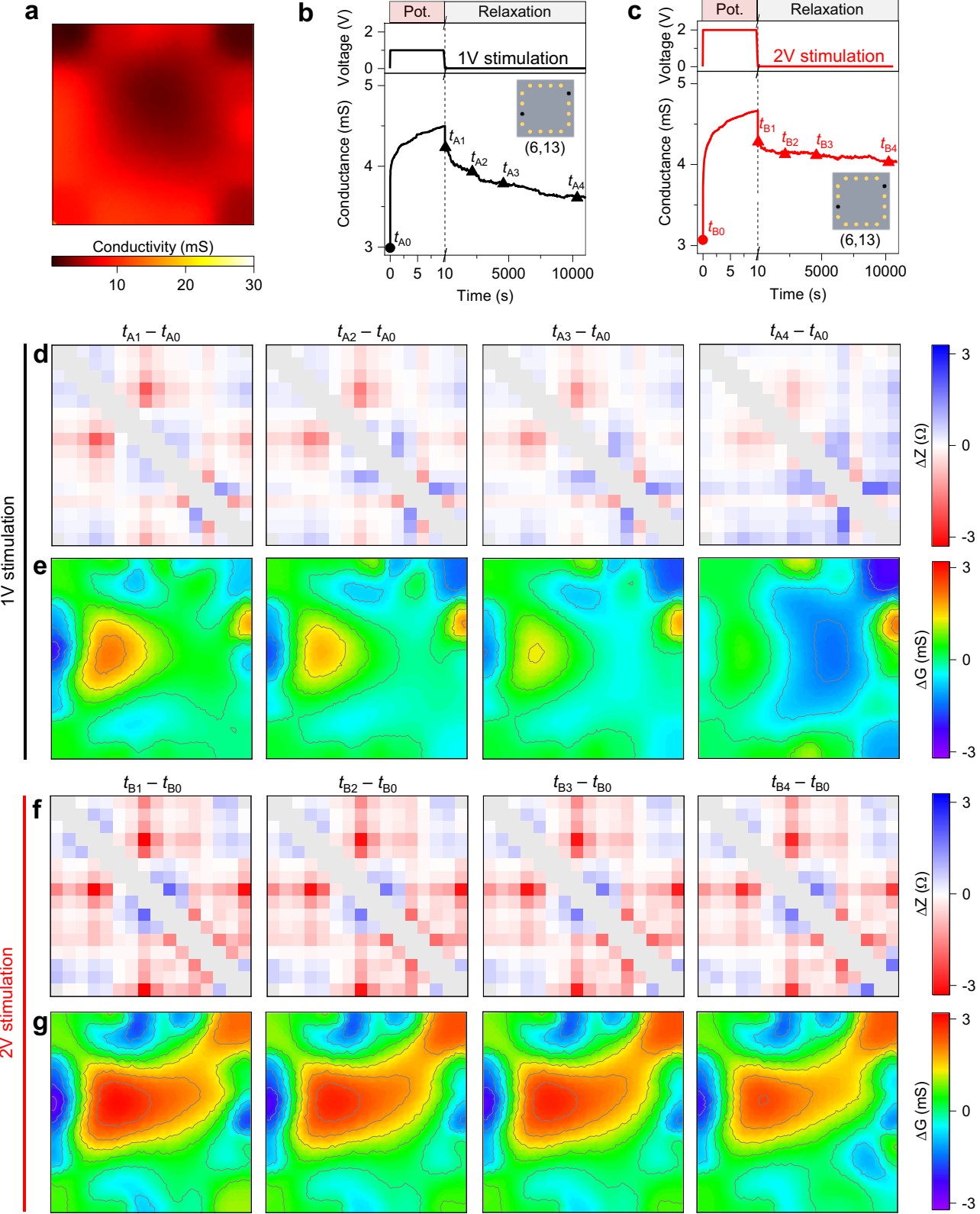

**Fig. 4 | Short-term and long-term memory effects in the NW network connectome. a** Experimental conductivity map of a non-homogenous NW network, showing a higher conductivity area at the bottom-left corner. Experimental evolution of the two-terminal effective conductance of the non-homogeneous network in between a pair of terminals under direct stimulation followed by spontaneous relaxation, where stimulation was with a 10 s voltage pulse of amplitude (**b**) 1 V and (**c**) 2 V. The relaxation was monitored over time in between the same terminals with a read voltage of 10 mV. Neuron terminals selected for stimulation (6 and 13) are highlighted in insets of (**b**, **c**). **d** Experimental differential impedance matrices and

(**e**) corresponding reconstructed conductivity maps of the non-homogeneous connectome after 1 V pulse stimulation, showing a progressive vanishing of the activation pattern induced by stimulation due to short-term memory effects. **f** Experimental differential impedance matrices and (**g**) corresponding reconstructed conductivity maps of the non-homogeneous connectome after 2 V pulse stimulation reported in (**c**) showing long-lasting changes in the connectome due to long-term memory effects. Differential conductivity maps show that the activation pattern is inherently related to the pristine conductivity map of the non-homogeneous network.

## Discussion

Results show that the emergent behavior of the NW network connectome arises from stimulation-induced spatio-temporal activation patterns related to the interplay in between the spatial location of stimulation, its temporal sequence, and the network topology. The internal memristive dynamics of the network connectome can be inferred through the temporal evolution of the impedance matrix of the system by means of multiterminal ERT mapping. Differently from other techniques for direct visualization of conductive pathways in self-organizing systems, ERT provides quantitative information on conductivity changes across the entire neuromorphic network (a comparison of measurand, spatial resolution, scanning area and acquisition time of ERT with other characterization techniques is reported in Supplementary Table 2). This is in analogy to brain mapping techniques that provide information on the connectivity of brain areas, without single synapse level resolution (a comparison with the effective resolution of brain mapping techniques is reported in Supplementary Table 3). In this context, is it worth mentioning that an approach based on ERT can be explored in a wide range of multiterminal neuromorphic devices with different number and position of contacts, where both spatial and temporal resolution can be improved by further optimizing measurement protocols and reconstruction algorithms (Supplementary Note 10).

In agreement with grid-graph modeling, the emergent behavior of high-density and nearly homogeneous NW networks shows memory traces that, once appropriately stimulated, progressively restore the initial spatial distribution of conductivity across the network connectome. Instead, a topology-related transition from short-term to long-lasting changes in the network connectome can be induced by means of appropriate stimulations, as revealed by considering non-homogeneous networks. The emergent behavior of the network results from the interplay in between (i) the peculiar spatio-temporal distribution of the electrical potential during stimulation—that depends on the initial conductivity map (i.e., on the network topology) that locally drives switching events—and (ii) the different switching properties of network areas with different local densities of NWs. Long-lasting activation patterns are expected in areas that experience enhanced voltage drops that can cause both (i) the formation of non-volatile filaments in highly stimulated memristive network elements, and (ii) voltage-induced local structural changes in the network topology (wiring or structural plasticity)[21].

In this context, the functional synaptic conductivity map of the NW connectome represents a memory state that depends on the history of spatial and temporal sequences of stimulation. The co-existence of long-term memory and memorizing-forgetting effects paves the way to the imprinting of cognitive information on a physical substrate in the form of artificial engrams, thus emulating biological engrams that endow the representation of experience stored in the brain[11]. Such a feature can represent a new paradigm for physical reservoir computing, since only short-term memory effects were currently exploited[49,50] (for this purpose, the repeatability of network outputs under the same input was demonstrated in NW networks operating in the short-term regime in ref. 36). Besides, spatially distributed short-term and long-term engram memory in the same physical substrate can lead to new unconventional computing paradigms able to process time-dependent information through short-term memory, while learning from experience and storing information thanks to long-lasting engram memory. These computing paradigms can explore the dependence of the non-linear dynamic evolution of the network connectome where activation patterns rely on the spatial location and temporal sequence of multiple input signals. In perspective, this can enable the realization of low-cost intelligent systems able to interact with the environment by receiving and responding to external stimulations, adapting their internal functional connectivity map to enable distribution, processing, and storage of information. In this framework, our results suggest that the network topology can be tailored to control emergent functionalities of self-organizing system, thus representing a key aspect for the hardware-software codesign of neuromorphic chips based on self-assembled nanonetworks. Furthermore, the knowledge over time of the conductivity map can be exploited, in perspective, as electrical transfer function to model the system's outputs for each possible input. This allows not only to quantitatively predict the input/output relation of arbitrarily placed input/output contacts, but also provides information on the electric field and current density distribution over the network under arbitrary external electrical stimulations. Therefore, the integration of the electrical transfer function of these multiterminal devices in circuit simulators and control systems can represent a turning point for the optimized design, realization and programming of neuromorphic chips based on self-organizing nanoarchitectures.

Finally, such a solid-state device would represent an alternative platform for neuroscientists to implement their new theoretical hypotheses about how memory is formed and recalled in engrams, and how memory is involved with learning in the formation of knowledge. The reverse-engineering of how NW networks emulate synaptic functionalities and how their topology affects the computational capabilities and engram representations may ultimately give new insights for understanding how biological brain networks work, retargeting the original goal of neuromorphic electronics.

## Methods

### Ag NW network fabrication

Memristive NW networks were realized by means of a drop-casting technique[21,36], by using Ag NWs with a diameter of 115 nm and length of 20–50 μm in isopropyl suspension (from Sigma-Aldrich) on a $10 \times 10$ mm² quartz substrate. In this work, the concentration of NWs in suspension and the drop volume was controlled to realize NW networks with an areal mass density (AMD) in the range ~99–136 mg m⁻². The NW network morphology was characterized by means of scanning electron microscopy (SEM; FEI Inspect F). Chemical and structural characterization of Ag NWs are reported in our previous work[21], showing the presence of a PVP insulating shell layer of ~1–2 nm surrounding the NW core. Besides its role as active material for resistive switching, the PVP shell layer prevents direct contact of the Ag inner core with the surrounding atmosphere, contributing to its chemical stability.

### Experimental setup for multiterminal characterization

Multiterminal electrical measurements were performed by means of a Keysight 34980A multifunction unit loaded with a Keysight 34933 switch matrix module, a Keithley 2602B source-meter and an Agilent 34461A digital multimeter. Multiterminal electrical characterization were performed by contacting the samples with spring-mounted needle probes through a custom fixture, as described in previous works (details in Supplementary Fig S2)[45,46,51]. Needle probes have a contact section of about 40 μm in diameter, ensuring reliable contacts to NW networks with AMD falling within the range 60–181 mg/m²[46]. The impedance matrix, electrically describing the NW network internal state, was acquired according to the so-called adjacent measurement protocol with constant voltage excitation that maximizes the signal-to-noise ratio in the measurements, while preventing electrical alterations of the NW network sample, as detailed in our previous work (details in Supplementary Note 3)[45]. For each measurement configuration represented by a pair of adjacent source terminals $(i,j)$ and a pair of adjacent sense terminals $(k,l)$, the trans-resistance is calculated as

$$R_{i,j;k,l} = V_{\text{sense}; k,l} / I_{\text{source}; i,j} \qquad (1)$$

where $I_{\text{source; i,j}}$ is the measured current flowing in between adjacent source terminals when a voltage bias $V_{\text{source; i,j}}$ is applied, while $V_{\text{sense; k,l}}$ is the voltage drop measured across adjacent sense terminals. Measurements were performed in four-terminal configuration (transresistance) thus excluding the effect of contact resistance (i.e., measurements in two-terminal and three-terminal configurations where terminals are shared with the source and/or sense terminal pairs are not considered). For each measurement configuration $i,j; k, l$, the transresistance value was the average of the forward and the reverse measurement configuration (defined by reversing the source and sensing polarity) to minimize thermoelectric effects (stray measurement offset). A measurement pattern (208 configurations) was performed in about 40 s. Multiterminal stimulation of the memristive NW network was performed by sequentially applying voltage pulses in between different pairs of electrical contacts. When not stimulated, electrical contacts were left floating. Spatio-temporal evolution of memristive NW networks under electrical stimulation with a spatio-temporal pattern was monitored by sequentially reading the impedance matrix of the system over time, while sequentially also monitoring the evolution of the resistance in between pairs of selected terminals. All measurements were performed in ambient air at controlled room temperature ($23\,°C \pm 0.5\,°C$).

### Grid-graph modeling

The homogeneous NW network deposited on a $10 \times 10\ \text{mm}^2$ quartz substrate was modeled as a grid graph ($21 \times 21$) nodes with memristive edges (pixel size of $0.05\ \text{mm}$), where contact terminal nodes are positioned according to the experimental sample geometry (details in Supplementary Note 4). Spatial anisotropy effects were avoided by introducing randomly oriented memristive diagonal edges. The current flow is regulated by Kirchhoff's law, while memristive dynamics of graph edges is regulated by a physics-based potentiation-depression rate-balance equation (details in Supplementary Note 9)[36,47,48]. Grid-graph modeling was performed in Python by exploiting the `NetworkX` package.

### Tomographical map reconstruction

Conductivity maps of the NW network samples were obtained from the ERT multiterminal measurements by means of tomographical image reconstruction. ERT reconstruction is an ill-posed inverse problem prone to measurement noise in the input data that requires regularization. In our approach, the tomographic reconstruction through the solution of the inverse problem is based on the minimization of the regularized functional:

$$\sigma = \arg\min\left(||Rcalc(\varsigma) - R||^2 + \Omega||L(\varsigma)||^2\right) \qquad (2)$$

where R is the input vector, $Rcalc(\varsigma)$ is a vector of transresistances calculated by the solver using the guess conductivity distribution $\varsigma$, and $||\cdot||$ is the Euclidean norm operator. $\Omega$ is the regularization parameter, a scalar used to set the amount of regularization $||L(\varsigma)||$ computed using the `EIDORS`' "`Laplace_prior`" function $L$.

To solve the problem described by Eq. (2) we applied numerical methods, implementing a Gauss-Newton solver within `EIDORS` v3. The numerical solution of (2) was based on a Finite Element Model (FEM) of the sample, having 9554 elements, including information about the position of the 16 contacts, the excitation voltage value, and the measurements' sequence corresponding to the implemented adjacent measurement protocol. The FEM was generated as a `MATLAB` structure (Supplementary Fig S17), then passed to the `EIDORS` solver. The Gauss–Newton solver can be configured to work in both "static" and "differential" modes. Static mode allows to retrieve absolute maps from single sets on ERT multiterminal measurements, while differential mode allows to retrieve the map of the difference between two different ERT transresistance measurements acquired at different timesteps. In the case of differential maps both $R$ and $Rcalc(\varsigma)$ are defined as the difference between measured or calculated transresistances at two different timesteps.

The amount of regularization $\Omega$ was selected by minimizing the differential map between two nominally identical sets of ERT measurements obtained with the sample in the pristine state. In this way $\Omega$ was large enough to damp the measurement noise, while any substantial over smoothing of the solution was avoided. The FEM conductivity maps were exported on a square grid of $100 \times 100$ pixels, which were enough to avoid artefacts due to the interpolation with the FEM structure. Note that from the point of view of ERT, the large area NW network can be considered as a two-dimensional material without a defined thickness. In this context, the conductivity (local property of the network) and the conductance (property of the network defined at 2 or more terminals) are here represented in the same unit (S).

## Data availability
The data that support the findings of this study are available on Zenodo (https://doi.org/10.5281/zenodo.8208381).

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

## Acknowledgements

Part of this work was supported by the European project MEMQuD, code 20FUN06. This project (EMPIR 20FUN06 MEMQuD) has received funding from the EMPIR program co-financed by the Participating States and from the European Union's Horizon 2020 research and innovation program. Part of this work has been carried out at Nanofacility Piemonte INRiM, a laboratory supported by the "Compagnia di San Paolo" Foundation, and at the QR Laboratories, INRiM.

## Author contributions

G.M. and A.C. generated the idea and designed the experiments, performed device fabrication and characterizations, developed simulations, and analyzed the data. G.M., A.C., and C.R provided interpretation of the results. G.M, L.B, L.C., and C.R. provided funding and background laboratory capabilities. All authors participated in the discussion of results and revision of the manuscript.

## Competing interests

The authors declare no competing interests.
