## [Peer Review File · Nature Communications]

REVIEWER COMMENTS

Reviewer #1 (Remarks to the Author):

In this manuscript, the authors propose a novel approach using self-organizing memristive nanowire connectomes to create multi-terminal paradigm memristive neural networks. The authors employ a simple fabrication method to construct a memristive plane with a complex topological network, addressing key issues such as single/multiple stimuli, short-term/long-term memory, uniform/non-uniform plane, and simulation/measurement.

The research presented in this work demonstrates a strong foundation, clear thinking, and a deep understanding of the field. The results obtained are innovative and great potential for guiding the exploration of physical large-scale memristive networks. Therefore, I believe that this manuscript is suited for publication in this journal. However, I suggest that some revisions should be made to enhance the readability and comprehensibility of the article.

1. As a physical memristive device, it is necessary to supplement its long-term working stability test to verify whether it has a good working life.
2. It is necessary for the author to verify the repeatability and consistency of the corresponding output under the same input state.
3. It should be stated in the text what magnitude the current spatial resolution is.
4. Authors should check their language and simplify it for comprehension.

Reviewer #2 (Remarks to the Author):

The authors present important results and interpretation on the dynamic behavior induced in nanowire networks through analysis of multielectrode measurements on the responses of Ag nanowire networks to electrical stimulations. Regarding the methodologies/techniques experimentally adopted, I think there are not very new points except for the way of data processing. The experimental results are, however, meaningful and interesting ones suggesting an important process to consolidate long-term memory from short-term memory, and I believe it's worthwhile to consider publication in Nature Communications. I've encountered several problems and questions in the manuscript as described below.

The authors claim that static indirect representation of the formation of conductive paths have been analyzed at the nano/microscale from scanning electron microscopy[ref.30] or through thermographic images[ref.28]. The reference 30 is indeed observing static image of stable conductive pathways and it would be difficult to observe dynamic change of the conductive pathways, but the data shown in the reference 28 seems to be snapshots taken out from dynamic observation of conductive pathway formation and annihilation more directly with higher spatial resolution correlated to the actual network structure. In this sense, the tomographic and dynamical imaging seem to have been realized more directly using the thermographic imaging with lock-in technique than the method proposed by the authors.

Also, multielectrode analysis has been already reported for the nanowire networks by UCLA group from USA and the NIMS group from Japan, the latter is using the multielectrodes mainly for proving associative functions of neuromorphic networks though. Since the UCLA group is not really applying mathematical conversion from the electrical data into conductive pathway representation, and the authors' approach does provide a certain advances in the methodology to evaluate and, in the future, to utilize neuromorphic functionalities of nanowire networks for data processing and computing.

I believe the authors should consider the above issues and appropriately position the present research in the course of trends of the related activities in the world.

Another point is that, because of the grid-graph modeling, real-space representation of conductive pathways such as shown in Figure 3d and 3g, for example, is not representing actual pathway in the nanowire network. I expect the authors carefully describe this situation. Of course, in the case of uniform and dense network, the grid-graph modeling provides rather realistic real-space representation but how about the case of sparse network which exhibits small-world properties is my concern.

Reviewer #3 (Remarks to the Author):

[GRAMMAR AND STYLE: While the quality of English is relatively good, there are a number of issues to address:

1) There are too many uses of "Despite" in the introduction,

2) moreover, there are a few instances of incorrect tense both in the introduction and even the abstract; e.g., "Despite the emergent behavior was shown to rely". Perhaps, "previously shown" or "having been shown"]

TECHNICAL REVIEW:

This is a systematic and interesting application of ERT for the analysis of the conductivity dynamics in networks with emphasis towards neuromorphic computing.

While the work presented is indeed novel and provides clear observations of temporary and permanent conductivity within regions of a network, this phenomenon is not necessary new or unexpected. There have been a number of recent publications along the same lines, some already cited by the authors.

Therefore, there is an outstanding question about the additional insight or new knowledge that results from this work and the proposed methodology for the characterization of networks. Towards this end, it would be most relevant to include and contrast (perhaps in a table):

"RESOLUTION OF TECHNIQUE"

- 1) The number of Ag-nanowires in the network with a description of the estimation method.
- 2) The number of synapses (wire to wire interfaces) in the network with a description of the estimation method.

Contrast the above with, 3) the effective "pixel" area probed by the 16 electrode system, and 4) the effective "pixel" area used in the model.

This effective resolution should be then contrasted with the resolution of high volume data methods like fMRI and the parcellation used to group the data into brain regions.

This line of analysis would place the current method in perspective and emphasize or at least clarify its potential.

"APPLICABILITY OF FINDINGS"

While the conductivity maps presented are indeed the result of quantitative data from ERT that is consistent with simulations, the overall outcome is still rather qualitative; that is, the main outcome

appears to be that there are "areas" in the network that are permanently modified in terms of their "local" conductivity and that some areas are only temporarily changed. As such, it is not clear how this information can be used quantitatively or systematically to gain insight and perspective into how to design, program, or analyze the dynamics of large scale neural networks for neuromorphic computing. Unless this is clarified, despite the thoroughness and interesting results presented, the novelty and anticipated impact of this work is relatively low.

Point-by-point response

We thank the Reviewers for their positive assessments about our work and for the constructive and competent comments, which now give us the opportunity to substantially improve the quality of our manuscript. We here provide a point-by-point response addressing all the reviewers' comments supported by new experimental and simulation results. We have also extensively revised the main manuscript and supplementary information, accordingly. We believe that the revised version of our manuscript now complies with the high standard required for publication in *Nature Communications*.

Reviewer #1 (Remarks to the Author):

In this manuscript, the authors propose a novel approach using self-organizing memristive nanowire connectomes to create multi-terminal paradigm memristive neural networks. The authors employ a simple fabrication method to construct a memristive plane with a complex topological network, addressing key issues such as single/multiple stimuli, short-term/long-term memory, uniform/non-uniform plane, and simulation/measurement.

The research presented in this work demonstrates a strong foundation, clear thinking, and a deep understanding of the field. The results obtained are innovative and great potential for guiding the exploration of physical large-scale memristive networks. Therefore, I believe that this manuscript is suited for publication in this journal. However, I suggest that some revisions should be made to enhance the readability and comprehensibility of the article.

We thank the reviewer for the positive assessment of our work and for finding the manuscript suitable for publication in this journal. According to reviewer suggestions, we have performed new experiments to enhance the readability and comprehensibility of the article, as detailed below.

C1. As a physical memristive device, it is necessary to supplement its long-term working stability test to verify whether it has a good working life.

R1. According to the reviewer request, we have performed a new experiment to monitor the working life of the NW network. For this purpose, the transresistance pattern of a multiterminal NW network was monitored over time for 20 days, while the ERT setup was hermetically closed in a box to limit the interaction with the environment. The initial transresistance pattern of the network acquired after network fabrication is reported in Figure R1a, while Figure R1b reports the evolution over time of transresistance values of all 208 configurations. After a few initial days of stabilization (that can be probably related to the evaporation of residual solvent), transresistance values of the multiterminal NW network tend to stabilize to near constant values and no network failures were observed, showing good working life at least for 20 days.

In this context, it should be noticed that Ag NWs are surrounded by a polyvinylpyrrolidone (PVP) shell layer that results directly from the NW synthesis by means of polyol process, where this polymer is commonly used as surfactant to control the growth rates of selected planes of face-centered-cubic silver in order to obtain the NW aspect ratio [1][2]. The presence of this PVP capping layer contributes to the chemical stability of the Ag NW inner core, protecting its direct contact with the surrounding atmosphere. Despite this, we have shown in our previous work [3] that direct ambient exposure induces degradation of electrical properties of Ag NW networks in the timescale of days or weeks, as a consequence of sulfurization effects that are similar to the tarnishing effect reported in Ag bulk [4]. While a detailed analysis of the effect of atmosphere on neuromorphic Ag NW network will require further analysis and is out of the scope of this work, it should be pointed out that stability issues of

NW networks for the realization of neuromorphic devices can be properly mitigated by means of device encapsulation with a conformal passivation layer realized, for example, by atomic layer deposition (ALD), as discussed in the review of Sannicolo et al. [5].

Figure R1. Long-term stability of NW networks. **a.** Transresistance pattern of an Ag NW network after deposition and **b.** evolution over time of transresistance values of all the 208 configurations, monitored by placing the ERT setup in a hermetically closed box to limit the interaction with the environment. After initial stabilization, transresistance values of the multiterminal NW network tend to stabilize to near constant values and no network failures were observed after 20 days.

Changes in the manuscript:

We have included new experimental data on long-term stability of NW networks here reported in Figure R1 as a new supplementary figure (Supplementary Figure S12). A sentence recalling this point was added in the revised manuscript at page 11. Also, we have clarified in the methods section (page 24 of the revised manuscript) that, besides its role as active material for resistive switching, the PVP shell layer prevents direct contact of the Ag inner core with the surrounding atmosphere, contributing to its chemical stability.

C2. It is necessary for the author to verify the repeatability and consistency of the corresponding output under the same input state.

R2. We thank the reviewer for pointing out this aspect that was not sufficiently discussed in the previously submitted manuscript. For this purpose, we have analyzed: *i)* the repeatability and consistency of NW network dynamics, *ii)* the repeatability of the input in NW networks with nominally identical characteristics, and *iii)* the repeatability and consistency of the ERT measurement pattern, as discussed below.

i) Repeatability and consistency of output under the same input

While long-term synaptic plasticity effects in NW networks provides irreversible changes in the network conductivity map, as discussed in the manuscript, the repeatability of the output under the same input state in NW networks operating in the short-term plasticity regime represents an important aspect for the implementation of unconventional computing paradigms such as reservoir computing. In this context, the repeatability and consistency of the corresponding output under the same input state was deeply investigated in our previous work [6], where the dynamics of the NW network operating in the short-term regime were exploited for the implementation of physical reservoir computing. Here, the repeatability of electrical response of the network in multiterminal configuration was demonstrated by presenting 120 spatio-temporal stimulation patterns composed of train of pulses characterized by 4 timeframes contemporarily applied to 4 network contacts (30 stimulations for each pattern) and by monitoring the corresponding output over time (refer to Figure 3b of ref. [6] for details on experimental configuration). For the benefit of the reviewer, we report in Figure R2 an example of NW network outputs under stimulation with the same input pattern. Figure R2a reports the input pattern and corresponding voltage stimulation input in the form of pulse trains applied to different contacts. The NW-based circuit with 4 inputs/4 outputs is schematized in Figure R2b. Figure R2c reports the NW network 4 outputs under stimulation with pattern reported in Figure R2a repeated for 30 times (Figure R2d and e report details of the output waveforms). The good agreement of the corresponding output under the same input pattern reflects the repeatability of the dynamic response of networks operating in the short-term memory regime. Similar results were obtained also by considering different input patterns (details and additional data in ref. [6]), experimentally proving the reproducibility of the network response. In this context, we showed in ref. [6] that the good repeatability of the input/output dynamic response resulted in the high experimental accuracy in physical reservoir computing systems based on multiterminal NW networks. For the sake of completeness, our previous analysis of NW networks in 2-terminal configuration revealed also that *a)* the memristive behavior was maintained over cycling as tested by switching the device 300 times through full-sweep cycles (refer to Figure 2a of ref. [7]), and that *b)* potentiation through pulse trains can be cyclically induced after device relaxation (refer to Supplementary S9 of ref. [7]).

Figure R2. Repeatability and consistency of the network output under the same input pattern. **a.** Input pattern and corresponding stimulating input voltage pattern in the spatio-temporal domain. **b.** experimental configuration for NW network stimulation. **c.** Time traces of output voltages for 30 stimulations with the pattern reported in panel a. **d.** Detail of time traces reported in panel a with a focus on the output response waveform of the output when the corresponding input is directly stimulated. **e.** Detail of time traces reported in panel a with a focus on the output response waveform of the output when the corresponding input is not directly stimulated. Adapted from ref. [6].

ii) Repeatability of outputs under the same input in different NW networks

In addition, we have tested the repeatability of input/output dynamics in different NW networks. For this purpose, we have performed additional experiments to test the effect of an arbitrary stimulation voltage waveform in between arbitrary contacts on two NW networks with nominally identical characteristics (the test was performed with the ERT setup). Figure R3a and b report the stimulation voltage pattern and the corresponding evolution of the effective conductance for the two NW networks. Despite small deviations in conductance values, it can be observed that the two NW networks qualitatively show the same dynamics, characterized by potentiation during voltage stimulation followed by spontaneous relaxation after stimulation.

Figure R3. Repeatability of NW network dynamics in samples with nominally identical characteristics. Response of **a.** NW network 1 and **b.** NW network 2 of an arbitrary stimulation voltage waveform. Stimulation voltage waveforms, composed of two spaced voltage pulses with amplitude of 0.8 V and 3 V, respectively, and length of 10 s, were applied to selected contacts of two NW networks with nominally identical characteristics. Contacts selected for stimulation (6 and 15) are highlighted in insets. The effective conductance in between pulses was monitored by applying a read voltage of 10 mV.

iii) Repeatability of the ERT measurement pattern

For the sake of completeness, we provide additional data also on the repeatability and consistency of the ERT output pattern obtained with the ERT measurement protocol as input. Figure R4 reports the mean and standard deviation of the ERT pattern evaluated by repeating the ERT measurement protocol 10 times on the same NW network in the pristine state. A standard deviation in transresistance values $< 0.5\%$ was observed, showing that the measurement protocol maximizes the signal to noise ratio, while preventing the onset of sample alterations.

In this framework, it is worth noticing that the metrological control of the ERT technique is discussed in detail in ref. [8] (sec. 4), which is the basis of the IEC Technical Specification 62607-6-7:2023 [9]. The individual transresistance measurements are traceable to the SI through the periodic calibration of the electrical instruments involved; the uncertainty is better than 1%. The accuracy of the conductivity values given by the reconstructed maps has been validated, also by comparison with other measurement techniques [10], [11], on reference samples.

Figure R4. Mean and standard deviation ($\times 100$) of measured ERT patterns evaluated by repeating the ERT measurement protocol 10 times on a NW network in the pristine state. A standard deviation in transresistance values $< 0.5\%$ was observed, showing that the measurement protocol maximizes the signal-to-noise ratio while preventing the onset of sample alterations.

Changes in the manuscript:

- We have clarified through the revised manuscript that the repeatability of network outputs under the same input was demonstrated in our previous work, by adding a sentence recalling this aspect in the discussion paragraph (page 20) and by adding the proper reference.

- New experimental data on the repeatability of outputs under the same input in different NW networks with nominally identical characteristics have been added in the revised manuscript, adding here reported Figure R3 as a new supplementary figure (Supplementary Figure S3). A sentence recalling this point was added in the revised manuscript (page 6).
- New experimental data on the repeatability of the ERT measurement pattern have been added in the revised manuscript, adding here reported Figure R4 as a new supplementary figure (Supplementary Figure S6). A sentence recalling this point was added in the revised manuscript (page 9). Further details on the traceability and validation of the ERT have been added in Supplementary Note 5.

C3. It should be stated in the text what magnitude the current spatial resolution is.

R3. The reviewer is right, this is an important feature that we forgot to detail. Despite the evaluation of the accuracy of the ERT map and, consequently, the definition of the pixel size, still represents an open problem [12], the spatial resolution of our 16-contacts ERT setup can be considered in practice of the order of the inter-electrode distance (≈ 2 mm), as detailed below.

The definition of an effective pixel size depends on the sensitivity with respect to the input data (the four-terminal resistance measurements) and the specific chosen inverse-solver algorithm. In order to test the spatial resolution of the ERT setup exploited for NW network mapping, we investigated a representative case by performing ERT mapping on a fluorinated tin oxide (FTO) thin film. To experimentally check the spatial resolution related to the 16-electrode setup and reconstruction algorithm, the FTO sample, initially of uniform conductivity of 150 mS [10], had later been damaged with a thin linear cut (red marker in Figure R5a). The conductivity dip in Fig. R5a has a full width at half maximum (measured over the sample diagonal, see Figure R5b) of ≈ 1.7 mm. This value is comparable with the distance between two adjacent contacts of 2 mm (on each edge of the contacts array). Hence the effective-pixel lateral size of ≈ 2 mm can be considered reasonable in the present discussion. As detailed in Supplementary Note 11, the spatial resolution of the ERT can be improved by optimization of both measurement protocol and reconstruction algorithms.

Figure R5. Experimental investigation of 16-electrode ERT spatial resolution. **a.** Fluorinated tin oxide (FTO) thin film with a thin linear cut (white marker of about 50 μm). Before the linear cut, the FTO sample was characterized by a uniform conductivity of 150 mS [10]. **b.** Conductivity dip observed in the diagonal conductivity profile of the sample (in the direction of the black arrow in the map of panel a), where the full width at high maximum corresponds to ≈ 1.7 mm.

Changes in the manuscript: We have reported in the revised manuscript (page 10) the spatial resolution of the ERT setup used in this work. New experimental results concerning the experimental investigation of the spatial resolution of the ERT setup (here reported in Figure R5) were added as a new supplementary figure (Supplementary Figure S9) and the here reported discussion on spatial resolution of the ERT setup was added in the revised Supplementary Note 5.

C4. Authors should check their language and simplify it for comprehension.

R4. According to the reviewer's suggestion, we have carefully revised the language of the whole manuscript. To simplify readability and comprehensibility of the work, we have divided the text in subsections, we have expanded the discussion of results and we have provided additional details through the manuscript.

Reviewer #2 (Remarks to the Author):

The authors present important results and interpretation on the dynamic behavior induced in nanowire networks through analysis of multielectrode measurements on the responses of Ag nanowire networks to electrical stimulations. Regarding the methodologies/techniques experimentally adopted, I think there are not very new points except for the way of data processing. The experimental results are, however, meaningful and interesting ones suggesting an important process to consolidate long-term memory from short-term memory, and I believe it's worthwhile to consider publication in Nature Communications. I've encountered several problems and questions in the manuscript as described below.

We thank the reviewer for the positive assessment on our work and for finding the manuscript worthwhile of publication in Nature Communications. We are thankful for the constructive comments that gave us the possibility to improve our work.

C5. The authors claim that static indirect representation of the formation of conductive paths have been analyzed at the nano/microscale from scanning electron microscopy [ref.30] or through thermographic images [ref.28]. The reference 30 is indeed observing static image of stable conductive pathways and it would be difficult to observe dynamic change of the conductive pathways, but the data shown in the reference 28 seems to be snapshots taken out from dynamic observation of conductive pathway formation and annihilation more directly with higher spatial resolution correlated to the actual network structure. In this sense, the tomographic and dynamical imaging seem to have been realized more directly using the thermographic imaging with lock-in technique than the method proposed by the authors.

Also, multielectrode analysis has been already reported for the nanowire networks by UCLA group from USA and the NIMS group from Japan, the latter is using the multielectrodes mainly for proving associative functions of neuromorphic networks though. Since the UCLA group is not really applying mathematical conversion from the electrical data into conductive pathway representation, and the authors' approach does provide a certain advances in the methodology to evaluate and, in the future, to utilize neuromorphic functionalities of nanowire networks for data processing and computing.

I believe the authors should consider the above issues and appropriately position the present research in the course of trends of the related activities in the world.

R5. We agree with the reviewer that the text should be improved to appropriately position our research results in the state-of-the art of related activities.

Concerning the comparison of our approach with previously reported techniques, we agree with the reviewer that, while ref. [13] reports only static representations of the conductive paths through passive voltage contrast SEM images, thermographic imaging with lock-in technique reported in ref. [14] can provide dynamic imaging of conductive pathways. Although thermography allows to observe changes in the current distribution across the network after electrical stimulation, this technique does not provide any information on how the conductivity map of the network evolves after stimulation. Indeed, it provides qualitative information of the main current pathways within the network where most of the power is dissipated, where power dissipation depends on both local resistance and local flowing current, without giving information on the local conductivity of the network. In this framework, features in thermographic images emerge from both *i*) areas of higher NW density

characterized by higher thermal connectivity resulting in more radiation and/or *ii*) areas with higher resistance acting as bottlenecks that connect more conductive areas.

Differently from thermography where the measurand is represented by photons emitted by the material that are then used to infer local electrical properties of the network, the measurand in ERT is represented by the electrical response of the network. This allows direct and quantitative information of the conductivity distribution across the network. Note that, while thermography allows to visualize only main conductive pathways that dissipate more power, ERT allows to achieve information on the entire network, even in areas not directly subjected to stimulation. Also, while thermographic images in NW networks were acquired by stimulating the device with a voltage square wave with very high amplitude (150 V) [14], the measurement protocol developed for ERT mapping of NW networks does not imply sample stimulation with high voltages, allowing mapping the sample without altering the conductivity distribution (the measurement pattern for ERT reconstruction is acquired by applying 10 mV in between contacts). For the sake of completeness, we report in Table T1 a comparison of main features of the here proposed methodology with other characterization techniques.

Table R1. Comparison of ERT with other characterization techniques for direct visualization of conductive pathways in self-organizing nanonetworks.

Technique	Measurand	Resolution (Pixel size)	Scanning area	Acquisition time
ERT (this work)	Local conductivity	≈ 2 mm	≈ 1 × 1 cm ²	≈ 40 s (non-scanning technique)
Lock-in thermography [14]	Infrared emission (Current is dissipated within current transmitting pathways, IR intensity can be then converted to temperature)	3 μm (but the spatial resolution is several times worse than the pixel size because of the optical limit of the lens)	≈ 1 × 1 mm ²	50 s (non-scanning technique)
Scanning Electron Microscopy [13], [15]	Secondary electrons (Passive voltage contrast images, produced by a low energy electron beam of 2-4 kV)	≈ 10 nm*	up to ≈ 100 × 100 μm ²	ms – s* (scanning technique) [▲]
Conductive-AFM [15]	Local current (Current flowing in between the C-AFM tip and a reference electrode)	≈ few nm*	≈ 50 × 50 μm ²	≈ min* (scanning technique) [▲]

* Not specified, estimation based on technique properties.

▲ Acquisition time scales quadratically with the scanning area in scanning techniques

We are aware that colleagues from UCLA and NIMS already reported multielectrode characterization of neuromorphic networks in previous works [16]–[20]. However, in these works the multiterminal characterization was performed mainly to exploit the capability of the network to nonlinearly map an input pattern in an output signal in the context of reservoir computing or to emulate associative functions, without directly accessing the internal state of the NW network. In other words, previous works treated the NW network nearly as a black box that performs a transformation of the input.

Main achievements of our work beyond the state-of-the-art are:

i) a radically new approach based on electrical tomography for quantitative investigation of spatially distributed changes in the conductivity distribution of self-organizing neuromorphic networks, as discussed through a combined experimental and modeling approach by considering memristive NW networks. The novelty of the approach was highlighted also by reviewer #1 and #3.

ii) tomographical evidence of memory engrams (or memory traces) in self-assembled connectomes, i.e., chemical and physical changes in biological neural substrates supposed to endow the representation of experience stored in the brain, as revealed by the conversion of spatially correlated short-term plasticity effects in long-lasting engram memory patterns.

iii) experimental evidence of the inherent relationship between spatio-temporal activation patterns and network topology, as revealed by the dependence of the activation patterns on the initial conductivity map of the network.

Finally, it is important to remark that the knowledge of the conductivity map and its evolution over time of the NW network provides information on the internal state of the nanonetwork and its dynamics. The knowledge of the conductivity map allows to quantitatively predict the input/output relation over time of arbitrarily placed input/output contacts, providing the **electrical transfer function** of the multiterminal NW network that, in perspective, can be exploited to model the system's outputs for each possible input. The integration of the electrical transfer function in circuit simulators and control systems can represent a turning point for the optimized design, realization and programming of neuromorphic chips based on self-organizing nanoarchitectures.

Changes in the manuscript: We have revised the introduction paragraph to better position our research results in the state-of-the art of related activities and to better clarify main achievements of our work, adding proper additional references. A detailed comparison of ERT with other characterization techniques for direct visualization of conductive pathways in self-organizing nanonetworks (here reported in Table R1) have been added as a new supplementary table (Supplementary Table 2). A sentence recalling this table has been added in the discussion paragraph (page 19 and 21).

C6. Another point is that, because of the grid-graph modeling, real-space representation of conductive pathways such as shown in Figure 3d and 3g, for example, is not representing actual pathway in the nanowire network. I expect the authors carefully describe this situation. Of course, in the case of uniform and dense network, the grid-graph modeling provides rather realistic real-space representation but how about the case of sparse network which exhibits small-world properties is my concern.

R6. We agree with the reviewer that the modeling approach based on grid-graph was not carefully described in the manuscript and should be described in more details.

The grid-graph modeling approach here exploited to model the emergent behavior of the homogeneous and high-density NW network relies on two main steps [21]: *i)* approximation of the NW network as a continuous medium, *ii)* parcellation of the 2D continuous domain and approximation as a regular grid graph.

Concerning step *i)*, previous theoretical and experimental investigation of NW networks by Forrò et al. [22] and Sannicolo et al. [23], respectively, revealed that the voltage distribution across a sufficiently dense NW network can be approximated to the voltage distribution observed in a continuous medium. This approximation was shown to hold already for NW network just above the percolation normalized densities $D > 2D_c$, where $D_c = 5.63$ is the percolation critical density [22]. This approximation is valid in our work where NW networks with $D > \approx 91$ were considered.

Concerning step *ii)*, the 2D plane representing the continuous material is parceled by means of a $N \times N$ mesh, and the pixelated network is represented through the grid-graph model built by associating a node to each pixel of the parceled domain and by introducing memristive edges to let neighbor nodes communicate (diagonal edges are introduced to make the network isotropic). A similar parceling of a continuous material was performed for modeling memristive materials and insulating layers of resistive switching cells [24], [25].

The aim of the grid-graph model is to catch main features of emergent memristive functionalities at the macroscale, as demonstrated in our previous works [6], [21], where the memristive behavior is decoupled from the particular and detailed behavior of each network element. In these terms, each memristive edge represents the memristive interaction in between different areas (nodes) of the network composed by a multitude of NWs. The consistency of this approach is supported also by the observation that the emergent memristive behavior of self-assembled networks can be described by an equation where complexity can be reabsorbed into the effective parameters of a single memristive element, as recently theoretically shown through a mean-field theory approach [26]. In these terms, results reported in Figure 3e (and corresponding maps in Figure 3g) show how network areas evolve under electrical stimulation. Results show that, at least in high-density networks, electrical stimulation results in the formation of an activation pattern that relies on the specific stimulation and is spatially distributed across the network, where conductance changes are not limited to the shortest path connecting the stimulated contacts. The occurrence of conductive pathways with multiple branches is in accordance also with experimental studies by passive voltage contrast SEM imaging reported in ref. [13].

In the case of sparse networks, we have performed additional simulations by considering a graph structure that reflects the NW network topology and its connectivity at the nano/microscale, as

reported in Figure R6. The network topology was modeled by randomly dispersing 1D objects on a 2D plane (Figure R6a), emulating the random dispersion of NWs on a substrate as described in refs. [27], [28] to realize a NW network with density $D \approx 2D_c$. In this case, the NW network was mapped into a graph where nodes and edges correspond directly to single NWs and NW junctions (Figure R6b). Electrical signal applied in between arbitrary selected source and ground nodes propagates through connected nodes that represent the electrical backbone of the system (Figure R6c). Figure 6d shows a direct visualization of the potential distribution in graph nodes when a voltage difference is applied in between source and ground nodes, while Figure 6e reports the voltage distribution across the 2D plane. As can be observed, already for NW densities $D \approx 2 D_c$, the voltage distribution resembles the voltage distribution expected in a continuous medium, in accordance with previous observations [22]. An example of activation pattern after network stimulation in between source and ground nodes is reported in Figure R6f. Also in this case, modeling shows that not only a single pathway, but multiple pathways emerge after stimulation. Details on activation patterns in sparse networks with small-world topology can be found in our previous work [28].

Despite modeling the emergent behavior of networks by considering the real network topology can be of great interest, mapping topology of NW network at the macroscale ($\sim \text{cm}^2$) with nanometric resolution, as required for direct graph representation, represents a challenge. Also, the high number of nodes and junctions related to the huge number of nano objects forming the network would require a high computational power for modeling the emergent behavior. Instead, the grid-graph modeling can emulate main features of the emergent behavior of nanonetworks with a simple personal computer in the timescale of few minutes/hours, representing a versatile model also for exploring unconventional computing implementations [21]. Last, it is worth noticing that a similar approach based on parcellation is exploited in neuroscience where, due to the difficulties of mapping the entire connectome at the single synapse level, the graph representation of the brain connectome relies on describing entire brain regions as nodes, where edges provide connections in between these regions [29].

Changes in the manuscript: The situation reported in Figure 3 was described in more detail at page 14 of the revised manuscript on the basis of the above reported discussion. Moreover, details on grid-graph modeling were added in the manuscript (page 9) and in the new Supplementary Note 4. Simulations of the emergent behavior of NW network at the micro/nanoscale were added as a new supplementary figure (Supplementary Figure S14) and a sentence recalling this point was added in the main manuscript (page 14).

Figure R6. Emergent behavior of the NW network connectome at the micro/nanoscale. **a.** NW network topology simulated by dispersing 1D objects (1800 NWs) on a 2D plane ($500 \times 500 \mu\text{m}^2$), where red dots represent NW midpoints while blue dots represent NW junctions, and **b.** corresponding graph representation. **c.** Electrical backbone of the network when stimulated in between source (red marked node, upper left) and ground node (black marked node, bottom right), **d.** corresponding visualization of the potential distribution across graph nodes when a voltage difference is applied between these nodes, and **e.** corresponding voltage distribution across the 2D plane. **f.** Activation pattern of the network after stimulation in between source and ground nodes with a voltage pulse, showing the emergence of a conductive pathway composed of multiple branches that connects source and ground nodes. Red intensity is proportional to edge conductance, blue intensity is proportional to node voltage.

Reviewer #3 (Remarks to the Author):

C7. [GRAMMAR AND STYLE: While the quality of English is relatively good, there are a number of issues to address:

- 1) There are too many uses of "Despite" in the introduction,
- 2) moreover, there are a few instances of incorrect tense both in the introduction and even the abstract; e.g., "Despite the emergent behavior was shown to rely". Perhaps, "previously shown" or "having been shown"]

R7. We thank the reviewer for pointing out these aspects. We have revised the entire manuscript to avoid repetition and to correct grammar mistakes.

C8. TECHNICAL REVIEW:

This is a systematic and interesting application of ERT for the analysis of the conductivity dynamics in networks with emphasis towards neuromorphic computing. While the work presented is indeed novel and provides clear observations of temporary and permanent conductivity within regions of a network, this phenomenon is not necessary new or unexpected. There have been a number of recent publications along the same lines, some already cited by the authors. Therefore, there is an outstanding question about the additional insight or new knowledge that results from this work and the proposed methodology for the characterization of networks. Towards this end, it would be most relevant to include and contrast (perhaps in a table):

"RESOLUTION OF TECHNIQUE"

- 1) The number of Ag-nanowires in the network with a description of the estimation method.
- 2) The number of synapses (wire to wire interfaces) in the network with a description of the estimation method.

Contrast the above with, 3) the effective "pixel" area probed by the 16 electrode system, and 4) the effective "pixel" are used in the model.

This effective resolution should be then contrasted with the resolution of high volume data methods like fMRI and the parcellation used to group the data into brain regions.

This line of analysis would place the current method in perspective and emphasize or at least clarify its potential.

R8. We thank the reviewer for the positive assessment on our approach of using ERT in this research field and for the comments on the resolution of the technique. We agree with the reviewer that a comparison of the proposed methodology with high volume data methods exploited for mapping the brain can provide additional insights for the reader, as detailed below.

In our work, we go beyond the state-of-the-art *i)* by proposing a radically new approach based on electrical tomography for quantitative investigation of spatially distributed changes in the conductivity of self-organizing neuromorphic networks, *ii)* by reporting tomographical evidence of memory engrams (or memory traces) in self-assembled connectomes, and *iii)* by reporting experimental evidence on the inherent relationship between spatio-temporal activation patterns and

network topology (detailed discussion of main achievements beyond the state-of-the art can be found in response R5 to reviewer 1).

Estimated NW density: $n \approx 5.7 \cdot 10^4 - 7.8 \cdot 10^4 \text{ mm}^{-2}$

This estimation was performed by considering the NW networks considered in our work, characterized by an areal mass density (AMD) in the range $\approx 99 - 136 \text{ mg m}^{-2}$, calculated by knowing the weight percentage of NWs in solution, the solution volume deposited on the substrate, the average NW dimensions, and the substrate surface area.

Estimated NW junction density: $j \approx 1.6 \cdot 10^6 - 3.1 \cdot 10^6 \text{ mm}^{-2}$

The estimation was performed through the formula $j = \frac{1}{2}P \cdot \pi n D = \frac{1}{2}P \pi n^2 L^2$ [28], where $P = 0.2027$ is the contact probability of 1D objects on a 2D plane (irrespective of the network density) [30], n is the NW density, D the normalized NW density and L the NW length. The calculated junction density is in line with previous estimation of $\approx 10^6$ NW junctions mm^{-2} from SEM imaging performed in a previous work [7].

The pixel size of the here proposed methodology performed with the 16-contacts system is $\approx 2 \text{ mm}$, corresponding to a synaptic junctions/pixel in the range $\approx 8 \cdot 10^6 - 1.2 \cdot 10^7$. A detailed discussion on the evaluation of the ERT resolution can be found in response R3 to Reviewer #1. The pixel size of the grid-graph modeling, where the $1 \times 1 \text{ cm}^2$ sample is mapped onto a grid of 21×21 nodes, is 0.05 mm , corresponding to a synaptic junctions/pixel in the range $\approx 4 \cdot 10^3 - 7.8 \cdot 10^3$. A comparison of the resolution of the ERT technique with techniques for brain mapping is reported in Table R2, together with an analysis of the synaptic junctions/pixel ratio.

Concerning brain parcellation, it must be noticed that different parcellation schemes can be adopted where, in the framework of graph theory, nodes can be defined as individual contacts of the electroencephalography or multielectrode-array, or as anatomically defined regions from imaging data [29].

Table R2. Comparison of the resolution of mapping techniques and synaptic densities in NW networks and human brain

NW network (2D)		
synaptic junction density: $\approx 2 \cdot 10^6 - 3 \cdot 10^6 \text{ mm}^{-2}$		
Technique	Pixel size	Synaptic junctions/pixel
ERT (this work)	$\approx 2 \text{ mm}$	$\approx 8 \cdot 10^6 - 1.2 \cdot 10^7$
Human brain (3D)		
synaptic density: $\approx 11 \cdot 10^8 \text{ mm}^{-3}$ * (ref. [31])		
Technique	Pixel size	Synaptic junctions/pixel
fMRI	$\approx 3 - 4 \text{ mm}$ pixel size of 500 um or less may be achieved with higher field magnets (7 T) (ref.[32])	$\approx 3 \cdot 10^{10} - 7 \cdot 10^{10}$
PET	$\approx 5 - 10 \text{ mm}$ limited by the size of the gamma-ray detectors as well as the positron-electron annihilation range (ref.[32])	$\approx 1 \cdot 10^{11} - 1 \cdot 10^{12}$
EEG	$\approx 10 - 20 \text{ mm}$ limited by the fact that unique reproduction of dipoles is not possible from scalp-based measurements and regularization should be employed for model estimation (ref.[32])	$\approx 1 \cdot 10^{12} - 9 \cdot 10^{12}$

*Average number of synaptic density in adult life

Changes in the manuscript: We have revised the introduction paragraph to better position our research results in the state-of-the art of related activities and to better clarify main achievements of our work. The estimated NW density and NW junction density were reported in the revised manuscript (page 6), while the estimation methods were described in a new supplementary note (Supplementary Note 1). We have reported a comparison of the resolution of mapping techniques and synaptic densities in NW networks and high-volume data methods for acquiring information on brain regions (here reported in Table R2) in a new supplementary table (Supplementary Table 3) and by adding a sentence recalling this point in the revised manuscript (page 19). The pixel size of the model has been added in the methods section.

C9. "APPLICABILITY OF FINDINGS"

While the conductivity maps presented are indeed the result of quantitative data from ERT that is consistent with simulations, the overall outcome is still rather qualitative; that is, the main outcome appears to be that there are "areas" in the network that are permanently modified in terms of their "local" conductivity and that some areas are only temporarily changed. As such, it is not clear how this information can be used quantitatively or systematically to gain insight and perspective into how to design, program, or analyze the dynamics of large scale neural networks for neuromorphic computing. Unless this is clarified, despite the thoroughness and interesting results presented, the novelty and anticipated impact of this work is relatively low.

R9. We agree with the reviewer that we did not explain in detail how information extracted by this technique could be quantitatively and systematically used for better neuromorphic architectures based on NW networks.

The main advantage of obtaining the conductivity map of the network in the framework of neuromorphic computing relies on the possibility, in perspective, of **exploiting the conductivity map as an electrical transfer function** that can model the system's outputs for each possible input. Indeed, the knowledge of the conductivity map (and its evolution over time) provides quantitative information of the internal state of the nanonetwork and its evolution. This knowledge allows not only to quantitatively predict the input/output relation of arbitrarily placed input/output contacts, but also provides information on the electric field and current density distribution over the network under arbitrary external electrical stimulations. The integration of the electrical transfer function in circuit simulators and control systems can represent a turning point for the optimized design, realization and programming of neuromorphic chips based on self-organizing nanoarchitectures. Complementing previous approaches where deep neural networks were exploited to model input-output characteristics of disordered networks for realizing functionalities [33], electrical transfer function obtained by ERT can in perspective provide a tool to efficiently optimize complex, multiterminal nanoelectronic devices based on self-assembled nanonetwork for desired functionality.

Changes in the manuscript: The above discussion was included in the discussion paragraph of the revised manuscript (page 20-21) to clarify how this approach could be exploited for the design of large networks for neuromorphic computing.

References

- [1] Y. Sun and Y. Xia, 'Large-Scale Synthesis of Uniform Silver Nanowires Through a Soft, Self-Seeding, Polyol Process', *Advanced Materials*, vol. 14, no. 11, p. 833, Jun. 2002, doi: 10.1002/1521-4095(20020605)14:11<833::AID-ADMA833>3.0.CO;2-K.
- [2] Y. Sun, B. Gates, B. Mayers, and Y. Xia, 'Crystalline Silver Nanowires by Soft Solution Processing', *Nano Lett*, vol. 2, no. 2, pp. 165–168, Feb. 2002, doi: 10.1021/nl010093y.
- [3] G. Milano *et al.*, 'Mapping Time-Dependent Conductivity of Metallic Nanowire Networks by Electrical Resistance Tomography toward Transparent Conductive Materials', *ACS Appl Nano Mater*, p. acsanm.0c02204, Oct. 2020, doi: 10.1021/acsnm.0c02204.
- [4] J. L. Elechiguerra, L. Larios-Lopez, C. Liu, D. Garcia-Gutierrez, A. Camacho-Bragado, and M. J. Yacaman, 'Corrosion at the Nanoscale: The Case of Silver Nanowires and Nanoparticles', *Chemistry of Materials*, vol. 17, no. 24, pp. 6042–6052, Nov. 2005, doi: 10.1021/cm051532n.
- [5] T. Sannicolo, M. Lagrange, A. Cabos, C. Celle, J.-P. Simonato, and D. Bellet, 'Metallic Nanowire-Based Transparent Electrodes for Next Generation Flexible Devices: a Review', *Small*, vol. 12, no. 44, pp. 6052–6075, Nov. 2016, doi: 10.1002/sml.201602581.
- [6] G. Milano *et al.*, 'In materia reservoir computing with a fully memristive architecture based on self-organizing nanowire networks', *Nat Mater*, vol. 21, no. 2, pp. 195–202, Feb. 2022, doi: 10.1038/s41563-021-01099-9.
- [7] G. Milano *et al.*, 'Brain-Inspired Structural Plasticity through Reweighting and Rewiring in Multi-Terminal Self-Organizing Memristive Nanowire Networks', *Advanced Intelligent Systems*, vol. 2, no. 8, p. 2000096, Aug. 2020, doi: 10.1002/aisy.202000096.
- [8] A. Cultrera and A. Catanzaro, 'Good Practice Guide on the electrical characterisation of graphene using noncontact and high throughput methods consortium', 2020. Accessed: Jun. 14, 2023. [Online]. Available: <http://empir.npl.co.uk/grace/>
- [9] IEC/TC 113 - International Electrotechnical Commission, 'IEC TS 62607-6-7:2023, "Nanomanufacturing - Key control characteristics - Part 6-7: Graphene - Sheet resistance: van der Pauw method"'. Accessed: Jun. 14, 2023. [Online]. Available: <https://webstore.iec.ch/publication/67057>
- [10] A. Cultrera *et al.*, 'Mapping the conductivity of graphene with Electrical Resistance Tomography', *Sci Rep*, vol. 9, no. 1, p. 10655, Dec. 2019, doi: 10.1038/s41598-019-46713-8.
- [11] A. Cultrera and L. Callegaro, 'Electrical Resistance Tomography of Conductive Thin Films', *IEEE Trans Instrum Meas*, vol. 65, no. 9, pp. 2101–2107, Sep. 2016, doi: 10.1109/TIM.2016.2570127.
- [12] V. Chitturi and N. Farrukh, 'Spatial resolution in electrical impedance tomography: A topical review', *J Electr Bioimpedance*, vol. 8, no. 1, pp. 66–78, Aug. 2019, doi: 10.5617/jeb.3350.
- [13] H. G. Manning *et al.*, 'Emergence of winner-takes-all connectivity paths in random nanowire networks', *Nat Commun*, vol. 9, no. 1, p. 3219, Dec. 2018, doi: 10.1038/s41467-018-05517-6.
- [14] Q. Li *et al.*, 'Dynamic Electrical Pathway Tuning in Neuromorphic Nanowire Networks', *Adv Funct Mater*, vol. 30, no. 43, p. 2003679, Oct. 2020, doi: 10.1002/adfm.202003679.
- [15] P. N. Nirmalraj *et al.*, 'Manipulating Connectivity and Electrical Conductivity in Metallic Nanowire Networks', *Nano Lett*, vol. 12, no. 11, pp. 5966–5971, Nov. 2012, doi: 10.1021/nl303416h.

- [16] H. O. Sillin *et al.*, 'A theoretical and experimental study of neuromorphic atomic switch networks for reservoir computing', *Nanotechnology*, vol. 24, no. 38, Sep. 2013, doi: 10.1088/0957-4484/24/38/384004.
- [17] A. V. Avizienis *et al.*, 'Neuromorphic atomic switch networks', *PLoS One*, vol. 7, no. 8, Aug. 2012, doi: 10.1371/journal.pone.0042772.
- [18] E. C. Demis *et al.*, 'Atomic switch networks—nanoarchitectonic design of a complex system for natural computing', p. 204003, May 2015, doi: 10.1088/0957-4484/26/20/204003.
- [19] S. Lilak *et al.*, 'Spoken Digit Classification by In-Materio Reservoir Computing With Neuromorphic Atomic Switch Networks', *Frontiers in Nanotechnology*, vol. 3, no. May, pp. 1–11, May 2021, doi: 10.3389/fnano.2021.675792.
- [20] A. Diaz-Alvarez, R. Higuchi, Q. Li, Y. Shingaya, and T. Nakayama, 'Associative routing through neuromorphic nanowire networks', *AIP Adv*, vol. 10, no. 2, 2020, doi: 10.1063/1.5140579.
- [21] K. Montano, G. Milano, and C. Ricciardi, 'Grid-graph modeling of emergent neuromorphic dynamics and heterosynaptic plasticity in memristive nanonetworks', *Neuromorphic Computing and Engineering*, pp. 0–22, Jan. 2022, doi: 10.1088/2634-4386/ac4d86.
- [22] C. Forró, L. Demkó, S. Weydert, J. Vörös, and K. Tybrandt, 'Predictive Model for the Electrical Transport within Nanowire Networks', *ACS Nano*, vol. 12, no. 11, pp. 11080–11087, Nov. 2018, doi: 10.1021/acsnano.8b05406.
- [23] T. Sannicolo *et al.*, 'Electrical Mapping of Silver Nanowire Networks: A Versatile Tool for Imaging Network Homogeneity and Degradation Dynamics during Failure', *ACS Nano*, vol. 12, no. 5, pp. 4648–4659, May 2018, doi: 10.1021/acsnano.8b01242.
- [24] E. Nedaaee Oskoe and M. Sahimi, 'Electric currents in networks of interconnected memristors', *Phys Rev E*, vol. 83, no. 3, p. 031105, Mar. 2011, doi: 10.1103/PhysRevE.83.031105.
- [25] Q. Li, A. Khiat, I. Salaoru, H. Xu, and T. Prodromakis, 'Origin of stochastic resistive switching in devices with phenomenologically identical initial states', *Proceedings - IEEE International Symposium on Circuits and Systems*, pp. 1428–1431, 2014, doi: 10.1109/ISCAS.2014.6865413.
- [26] F. Caravelli, G. Milano, C. Ricciardi, Z. Kuncic, G. Milano, and Z. Kuncic, 'Mean Field Theory of Self-Organizing Memristive Connectomes', 2023, doi: 10.1002/andp.202300090.
- [27] A. Loeffler *et al.*, 'Topological Properties of Neuromorphic Nanowire Networks', *Front Neurosci*, vol. 14, Mar. 2020, doi: 10.3389/fnins.2020.00184.
- [28] G. Milano, E. Miranda, and C. Ricciardi, 'Connectome of memristive nanowire networks through graph theory', *Neural Networks*, vol. 150, pp. 137–148, 2022, doi: 10.5281/zenod.
- [29] E. Bullmore and O. Sporns, 'Complex brain networks: graph theoretical analysis of structural and functional systems', *Nat Rev Neurosci*, vol. 10, no. 3, pp. 186–198, Mar. 2009, doi: 10.1038/nrn2575.
- [30] J. Heitz, Y. Leroy, L. Hébrard, and C. Lallement, 'Theoretical characterization of the topology of connected carbon nanotubes in random networks', *Nanotechnology*, vol. 22, no. 34, p. 345703, Aug. 2011, doi: 10.1088/0957-4484/22/34/345703.
- [31] H. Peter R., 'Synaptic density in human frontal cortex — Developmental changes and effects of aging', *Brain Res*, vol. 163, no. 2, pp. 195–205, Mar. 1979, doi: 10.1016/0006-8993(79)90349-4.

- [32] G. H. Glover, 'Overview of functional magnetic resonance imaging', *Neurosurgery Clinics of North America*, vol. 22, no. 2. pp. 133–139, Apr. 2011. doi: 10.1016/j.nec.2010.11.001.
- [33] H. C. Ruiz Euler *et al.*, 'A deep-learning approach to realizing functionality in nanoelectronic devices', *Nat Nanotechnol*, vol. 15, no. 12, pp. 992–998, Dec. 2020, doi: 10.1038/s41565-020-00779-y.

REVIEWERS' COMMENTS

Reviewer #1 (Remarks to the Author):

In this manuscript, the authors propose a novel approach using self-organizing memristive nanowire connectomes to create multi-terminal paradigm memristive neural networks. The authors employ a simple fabrication method to construct a memristive plane with a complex topological network, addressing key issues such as single/multiple stimuli, short-term/long-term memory, uniform/non-uniform plane, and simulation/measurement.

The research presented in this work demonstrates a strong foundation, clear thinking, and a deep understanding of the field. The results obtained are innovative and great potential for guiding the exploration of physical large-scale memristive networks. After completing some minor modifications, I believe that this manuscript is well-suited for publication in this journal.

Reviewer #2 (Remarks to the Author):

I believe that the revised manuscript is now sufficiently provide information to follow up the value of the work which leads research in this field. Dynamics is no matter what a key to understand working principles and emergent behavior of complex neuromorphic nanowire network. The authors provide an important insight to step forward the understanding.

Therefore, I believe the paper be valuable piece to be published in Nature communications.

Reviewer #3 (Remarks to the Author):

The authors has addressed all my comments and the modifications to the manuscript are consistent with the reply to the comments. The perceived impact of the work is nonetheless still low but the implications of that is for the Editors to decide.

Point-by-point response

We thank the Reviewers for the positive assessment of our work and for finding the revised version of the manuscript suitable for publication in *Nature Communications*. We acknowledge them for the constructive and competent comments, which gave us the opportunity to substantially increase the quality of our manuscript through the review process. We have revised the manuscript according to editorial requests, changes are highlighted in blue.

Reviewer #1 (Remarks to the Author):

C1. In this manuscript, the authors propose a novel approach using self-organizing memristive nanowire connectomes to create multi-terminal paradigm memristive neural networks. The authors employ a simple fabrication method to construct a memristive plane with a complex topological network, addressing key issues such as single/multiple stimuli, short-term/long-term memory, uniform/non-uniform plane, and simulation/measurement. The research presented in this work demonstrates a strong foundation, clear thinking, and a deep understanding of the field. The results obtained are innovative and great potential for guiding the exploration of physical large-scale memristive networks. After completing some minor modifications, I believe that this manuscript is well-suited for publication in this journal.

R1. We thank the reviewer for the positive assessment of our work and for finding the revised version suitable for publication in *Nature Communications*. We have performed minor modifications in the new revised version of the manuscript to comply with Editorial requests.

Reviewer #2 (Remarks to the Author):

C2. I believe that the revised manuscript is now sufficiently provide information to follow up the value of the work which leads research in this field. Dynamics is no matter what a key to understand working principles and emergent behavior of complex neuromorphic nanowire network. The authors provide an important insight to step forward the understanding. Therefore, I believe the paper be valuable piece to be published in Nature communications.

R2. We thank the reviewer for the positive assessment of our work and for finding the revised version suitable for publication in *Nature Communications*

Reviewer #3 (Remarks to the Author):

C3. The authors has addressed all my comments and the modifications to the manuscript are consistent with the reply to the comments. The perceived impact of the work is nonetheless still low but the implications of that is for the Editors to decide.

R3. We thank the reviewer for the positive assessment of our additional work reported in the revised version of the manuscript. Concerning the impact, we believe that our work that reported for the first time on topographical evidence of memory engrams in self-assembled nanowire networks by analyzing internal dynamics of these systems with a new approach based on Electrical Resistance Tomography (ERT) comply with the high standards and scope of *Nature Communications*.